# On the implicit minimization of alternative loss functions when training deep networks

## Abstract

Understanding the implicit bias of optimization algorithms is important in order to improve generalization of neural networks. One approach to try to exploit such understanding would be to then make the bias explicit in the loss function. Conversely, an interesting approach to gain more insights into the implicit bias could be to study how different loss functions are being implicitly minimized when training the network. In this work, we concentrate our study on the inductive bias occurring when minimizing the cross-entropy loss with different batch sizes and learning rates. We investigate how three loss functions are being implicitly minimized during training. These three loss functions are the Hinge loss with different margins, the cross-entropy loss with different temperatures and a newly introduced Gcdf loss with different standard deviations. This Gcdf loss establishes a connection between a sharpness measure for the $0 - 1$ loss and margin based loss functions. We find that a common behavior is emerging for all the loss functions considered.

## 1 Introduction

In the last few years, deep learning has succeeded in establishing state of the art performances in a wide variety of tasks in fields like computer vision, natural language processing and bioinformatics (LeCun et al., 2015). Understanding when and how these networks generalize better is important to keep improving their performance. Many works starting mainly from Neyshabur et al. (2015), Zhang et al. (2017) and Keskar et al. (2017) hint to a rich interplay between regularization and the optimization process of learning the weights of the network. The idea is that a form of inductive bias can be realized implicitly by the optimization algorithm.

In this paper, we investigate the implicit bias induced from using different learning rates and batch sizes when minimizing the cross-entropy loss with SGD. A common theory is that more noise in the gradient bias the solution toward flatter minima (Keskar et al., 2017). We draw a connection between a particular measure of flatness and margin based loss functions[1].

Our contributions are the following:

1. A new loss function (Gcdf loss) that can be interpreted as a measure of flatness for the $0 - 1$ loss (for the top layer's weights of the network).

2. A methodology consisting in tracking alternative loss functions during training and comparing them for a given training loss value to try to uncover implicit biases in the optimization algorithm applied to varying the learning rate and batch size in SGD.

3. Experimental results on CIFAR10 and MNIST showing that larger learning rates and smaller batch sizes are better at implicitly minimizing the cross-entropy loss with larger temperature parameter, the hinge loss with larger margin parameter and the Gcdf loss with larger standard deviation parameter. At the opposite, smaller learning rates and larger batch sizes are better at implicitly minimizing the cross-entropy loss, the hinge loss and the Gcdf loss with smaller values of their respective parameter.

---

[1]The concept of margin has been link to generalization of deep networks; see for example Bartlett et al. (2017), Poggio et al. (2019) and Jiang et al. (2019)

We do not propose to modify optimization algorithms to try to improve large batch training but we instead try to offer new insights on how the solutions it produces are different from solutions resulting from small batch training (or larger learning rates). The hope is to eventually succeed at incorporating the inductive bias in the objective being optimized instead of relying on the implicit bias of the optimization algorithm. It is not yet clear to what extent this goal can be realized (and by what means[2]) and we certainly do not claim to be reaching it. We offer only a partial understanding of some of the differences between large batch training (or using small learning rates) and small batch training (or using large learning rates) through the behavior of alternative loss functions during training.

## 2 RELATED WORK

It was observed by Zhang et al. (2017) that deep networks can often obtain good results without explicit regularization even if they have the capacity to essentially memorize the training set. They hypothesized that SGD is probably acting as an implicit regularizer. Also, the earlier work of Neyshabur et al. (2015) brought forward the idea that optimization might be implicitly biasing the trajectory toward low norm models. Since then, many works have investigated the idea of implicit regularization for neural networks (linear or non-linear). For example, Arora et al. (2019) studied how gradient descent finds low rank solutions for matrix completion with deep linear networks. Soudry et al. (2018) showed that gradient descent converges to the max-margin solution for logistic regression and Lyu & Li (2019) provides and extension to deep non-linear homogeneous networks. In contrast to these works, we study empirically how the optimization algorithm implicitly minimizes alternative loss functions during the course of training.

A highly studied source of implicit bias from the optimization algorithm is the ability to reach flatter minima. In Keskar et al. (2017), the worst loss that can be obtained when slightly perturbing the parameters is considered as a measure of sharpness while Neyshabur et al. (2017) considered the expected loss under Gaussian noise in the weights. We consider a measure of sharpness (section 3) similar to Neyshabur et al. (2017) and we apply it to the $0 - 1$ loss directly instead of the usual surrogate cross-entropy loss.

The batch size and the learning rate are two ways to control the noise in the gradient which might influence the sharpness of the resulting solution (see for example Smith & Le (2018), Smith et al. (2018)). In conjunction with increasing the learning rate, different strategies like training for more epochs (Hoffer et al., 2017), "warm up" (Goyal et al., 2017) and using a separate learning rate for each layer based on the norm of the weights (You et al., 2017) have been proposed to improve the performance of large batch training. Instead of trying to offer a new modification to the optimization algorithm, we try here to capture the inductive bias into computationally efficient to use loss functions in the hope of eventually simplifying the design of optimization algorithms.

## 3 GCDF LOSS

This section introduces a loss function based on the idea of flat minima. It is defined as a measure of sharpness for the $0 - 1$ loss. The main motivation for introducing this loss function is that it is simultaneously a measure of sharpness and a margin based loss function establishing a clear relationship between these ideas. Furthermore, as opposed to the cross-entropy loss and the Hinge loss, it is bounded and non-convex (see section 4.1 for a visual comparison). It thus offers more diversity to the loss functions investigated in this paper. We start with the binary linear case in 3.1 and then extend to the multi-class case in 3.2. For deep networks, this loss will be applied on the top layer. It is a possible extension to our work to consider loss functions applied on multiple layers maybe in a similar fashion to Elsayed et al. (2018).

---

[2]see for example Arora et al. (2019) about the difficulties to capture the implicit bias of gradient descent with norms.

### 3.1 BINARY LINEAR CASE

Let $f(w,x) = w^T x + b$, where $w, x \in \mathbb{R}^n$ and $b \in \mathbb{R}$. Consider the $0-1$ loss for a binary linear classifier: $L(f(w,x), y) = \mathbb{1}\left[y(w^T x + b) < 0\right]$, where $\mathbb{1}$ is the indicator function. Note that we will write all the loss functions for single examples $(x, y)$ throughout the paper and it will be understood that the training loss is obtained by taking the mean over the training set. We smooth (or "robustify") the $0-1$ loss by considering its expectation under Gaussian noise in the weights. This loss function will then be denoted by $L_\sigma(w, x, y)$ when the standard deviation is $\sigma$. Consider the random variable $\epsilon \sim \mathcal{N}(0, \sigma^2 I)$, where $\mathcal{N}(0, \sigma^2 I)$ is a zero mean isotropic Gaussian distribution with covariance matrix $\sigma^2 I$. Since $(w + \epsilon)^T x + b$ is distributed as a Gaussian distribution with mean $w^T x + b$ and variance $\sigma^2 ||x||^2$, we get that $y((w + \epsilon)^T x + b)$ is distributed as a Gaussian distribution with mean $y(w^T x + b)$ and the same variance. Therefore,

$$L_\sigma(w, x, y) = \mathbb{E}_\epsilon L(f(w + \epsilon, x), y) = \Phi\big(\frac{-y(w^T x + b)}{\sigma ||x||}\big),\tag{1}$$

where $\Phi$ is the Gaussian cumulative distribution function (Gcdf) given by

$$\Phi(z) = \frac{1}{\sqrt{2\pi}} \int_{-\infty}^{z} \exp\big(\frac{-t^2}{2}\big) dt.\tag{2}$$

If we assume that $x$ is normalized, the loss $L_\sigma$ is a (decreasing) function of $y f(w, x)$ (it is a margin based loss function in the terminology from Lin (2004) for example).

### 3.2 MULTI-CLASS CASE

Suppose the number of classes is $m$ and now consider the affine mapping $f(W, x) = Wx + b$ with $x \in \mathbb{R}^n$, $b \in \mathbb{R}^m$ and $W \in \mathbb{R}^{m \times n}$. For some fixed $x \in \mathbb{R}^n$ and denoting by $w_j$ the $j^{th}$ row of $W$, let $s_j := w_j^T x + b_j$ be the corresponding score for class $j$. Finally, let $s_j(\epsilon_j) := (w_j + \epsilon_j)^T x + b_j$ be the perturbed score, $\epsilon_j$ an isotropic Gaussian random variable with mean $0$ and covariance matrix $\sigma^2 I$. For a given class $y$, we get

$$P\big\{s_y(\epsilon_y) \neq \max_j s_j(\epsilon_j)\big\} \quad \leq \quad \sum_{j \neq y} P\big\{s_j(\epsilon_j) > s_y(\epsilon_y)\big\}\tag{3}$$

$$= \quad \sum_{j \neq y} P\big\{s_j - s_y > (\epsilon_y - \epsilon_j)^T x\big\}\tag{4}$$

$$= \quad \sum_{j \neq y} \Phi\left(\frac{s_j - s_y}{||x|| \sigma \sqrt{2}}\right),\tag{5}$$

since $(\epsilon_y - \epsilon_j)^T x$ follows a zero mean Gaussian distribution with variance $2\sigma^2 ||x||^2$. We define

$$L_\sigma(W, x, y) := \sum_{j \neq y} \Phi\left(\frac{s_j - s_y}{||x|| \sigma \sqrt{2}}\right).\tag{6}$$

This is an upper bound on the probability that the classifier does not predict $y$ under Gaussian noise on $W$. We will experiment with this Gcdf loss function on top of feedforward neural networks (and also with other loss functions) in the following sections. In all the experiments, we use normalization to enforce $||x|| = 1$ (this $x$ now represents the feature vector for the top layer).

## 4 IMPLICIT MINIMIZATION OF DIFFERENT LOSS FUNCTIONS

In this section, we track different loss functions while training deep neural networks with the cross-entropy loss varying the learning rates and batch sizes in SGD with momentum. The results in the main text are obtained while training on CIFAR10. Results on MNIST are given in Appendix A. The following loss functions are considered: cross-entropy with different values of temperature, Hinge loss with different margin parameters and the Gcdf loss with different standard deviation parameters.

For the cross-entropy loss, the temperature $T$ divides the scores $s_j$ before the softmax function. That is, the probability for class $j$ is then given by

$$\frac{\exp(s_j/T)}{\sum_k \exp(s_k/T)}. \tag{7}$$

Remark that the positive homogeneity of the Relu implies that normalizing each layer of the network is equivalent to take $T$ equal to the product of the norm of the layers. The cross-entropy loss after normalization at the end of training is investigated in Liao et al. (2018). In contrast, we consider here multiple values for $T$ and investigate the behavior during training. Given the probabilities for each class, the cross-entropy loss (on a single example) is then the negative $\log$ probability for the correct class. For its part, the multi-class Hinge loss with margin parameter $\gamma$ (on a single example) is given by

$$\sum_{j \neq y} \max\{0, \gamma + (s_j - s_y)\}. \tag{8}$$

The Gcdf loss with standard deviation parameter $\sigma$ has been described and motivated in the previous section.

## 4.1 Visual comparison of the loss functions

Assume that we have two classes and let $z = s_y - s_j$ (for $j \neq y$). An example is correctly classified if $z > 0$. The Gcdf loss is then given by $\Phi(\frac{-z}{\sigma\sqrt{2}})$, the Hinge loss by $\max\{0, \gamma - z\}$ and the cross-entropy loss by $\log(\exp(\frac{-z}{T})+1)$. These functions are plotted in figure 1. They share one interesting characteristic on the side $z > 0$: when their parameter ($\sigma$, $T$ or $\gamma$) gets larger, the loss takes more time to get closer to zero when $z$ increases. This kind of "heavier tail" behavior can encourage larger $z$ values for some training points at the expense of other closer to zero training points more easily.

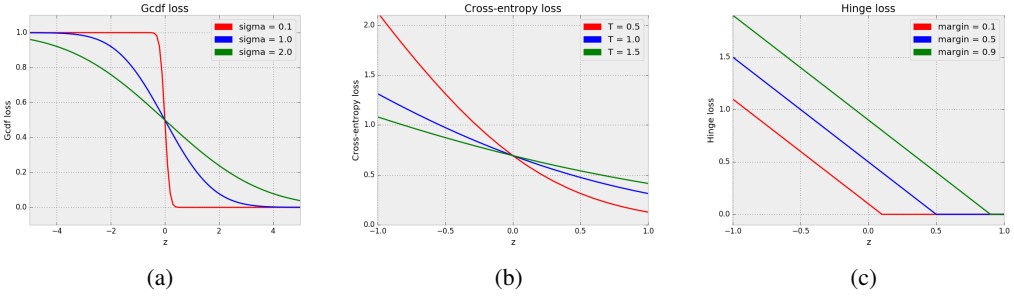

(a)           (b)           (c)

Figure 1: In (a), we have the Gcdf loss for different values of sigma. In (b), the Cross-entropy loss for different temperatures and in (c), the Hinge loss for different margins.

## 4.2 Training curves of the alternative loss functions

In figures 2, 3 and 4, we investigate the effect of the size of the learning rate by considering the implicit training curves of the Gcdf loss, the cross-entropy loss and the Hinge loss for different values of their respective parameter ($\sigma$, $T$ or $\gamma$). The learning rate is kept constant throughout training (no decaying schedule is used). We consider a small learning rate of $0.001$ and a larger learning rate of $0.1$. For the three loss functions considered, the larger learning rate is clearly better at implicitly minimizing them for larger values of their parameter. A similar conclusion holds when considering different batch sizes as is shown in figures 5, 13 (Appendix B) and 14 (Appendix B). In this case, the smaller batch size (256) is much better at implicitly minimizing the loss functions for larger values of their parameter than the larger batch size (16384). As a technical aside, note that ghost batch normalization Hoffer et al. (2017) is used when training with large batch sizes. The gradients are accumulated on a sequence of smaller mini-batches of size 256 before updating the weights.

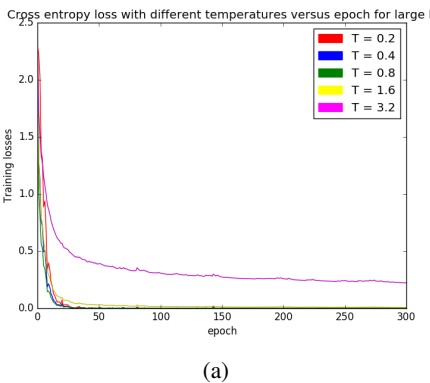 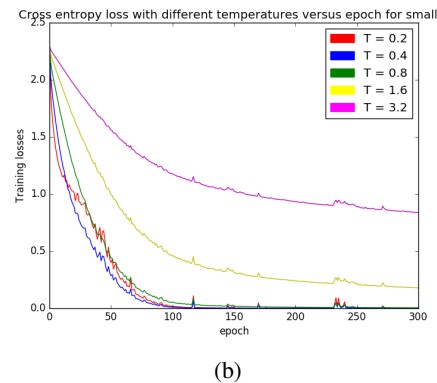

(a)                     (b)

Figure 2: Tracking the the cross-entropy loss with different temperatures while training with the standard cross-entropy loss ($T = 1$) on CIFAR10. A relatively large learning rate of $0.1$ is used in (a) while a much smaller learning rate of $0.001$ is used in (b). Even though both learning rates succeed at minimizing to almost zero the loss function they are trained on, the smaller learning rate does not implicitly minimize as well the cross-entropy loss for larger temperatures.

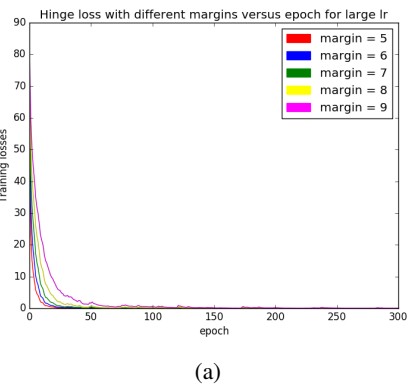 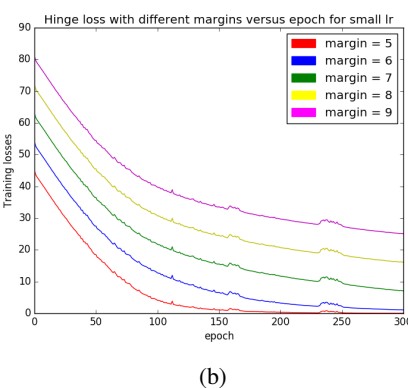

(a)                     (b)

Figure 3: Tracking the Hinge loss with different margin parameters while training with the standard cross-entropy loss on CIFAR10. A relatively large learning rate of $0.1$ is used in (a) while a much smaller learning rate of $0.001$ is used in (b). Even though both learning rates succeed at minimizing to almost zero the loss function they are trained on, the smaller learning rate does not implicitly minimize as well the Hinge loss for larger margins.

### 4.3 Alternative loss versus actual train loss

At a given fixed training loss value two training runs have made the same progress toward minimizing their objective function but they might not have made the same progress with respect to other measures of performance. The other measures of performance considered here are of course our alternative loss functions. In figure 6, we plot the Gcdf loss against the actual train loss for different runs corresponding to different learning rates. We can see that smaller learning rates are actually better at minimizing the Gcdf loss with smaller $\sigma$ during training while larger learning rates are better at minimizing the Gcdf loss with larger $\sigma$. There exists an intermediate value (here $\sigma = 1$) where all the learning rates considered in our experiments are essentially equivalent at implicitly minimizing the Gcdf loss. The train error behaves similarly to the Gcdf loss with a small value of $\sigma$. In the binary case, the Gcdf loss converges pointwise to the $0 - 1$ loss almost everywhere (except at 0) when $\sigma$ goes to 0. It would therefore make sense to actually define the Gcdf loss for $\sigma = 0$ to be the train error (in the binary case; some modifications are needed in the multi-class case). In this light, it is not surprising that figure 6a and 6d are showing a similar behavior. In order to make even more clear how different choices of learning rates are not implicitly minimizing the alternative loss functions for different parameters ($\sigma$, $T$ or $\gamma$) in the same way, we plotted the alternative losses

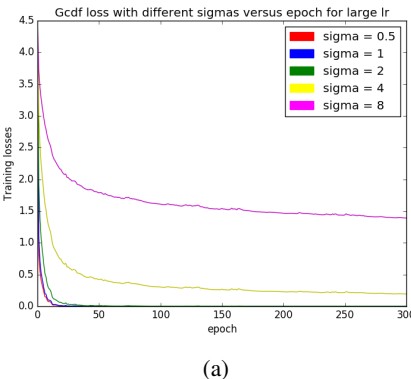 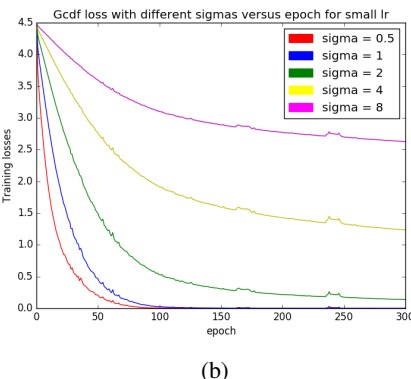

(a) (b)

Figure 4: Tracking the Gcdf loss with different standard deviation parameters while training with the standard cross-entropy loss on CIFAR10. A relatively large learning rate of $0.1$ is used in (a) while a much smaller learning rate of $0.001$ is used in (b). Even though both learning rates succeed at minimizing to almost zero the loss function they are trained on, the smaller learning rate does not implicitly minimize as well the Gcdf loss for larger standard deviations.

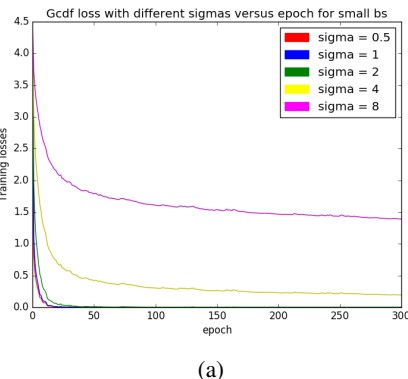 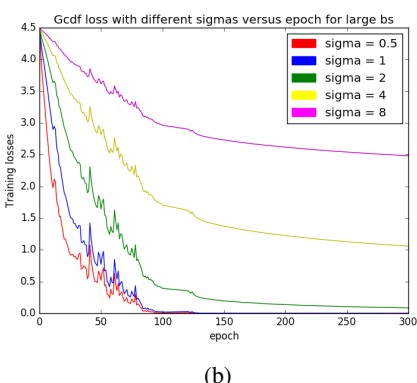

(a) (b)

Figure 5: Tracking the Gcdf loss with different standard deviation parameters while training with the standard cross-entropy loss on CIFAR10. A relatively small batch size of $256$ is used in (a) while a much larger batch size of $16384$ is used in (b). Even though both batch sizes succeed at minimizing to almost zero the loss function they are trained on, the larger batch size does not implicitly minimize as well the Gcdf loss for larger values of $\sigma$.

against their respective parameter for some fixed training loss value in figure 7. Similar results are obtained on MNIST (see figure 12 in Appendix A). See also figure 16 (Appendix B) for the results when considering different batch sizes on CIFAR10.

## 5 DISCUSSION AND CONCLUSION

Suppose $\epsilon$ is distributed according to an isotropic Gaussian distribution with covariance matrix $\sigma^2 I$ and mean $0$. Under a second order approximation to the cross-entropy loss $L_c(w, x, y)$ at $w$, we get $\mathbb{E}_\epsilon \left[ L_c(w + \epsilon, x, y) \right] \approx L_c(w, x, y) + \frac{\sigma^2}{2} \operatorname{Tr}(H)$, where $H$ is the Hessian of $L_c(w, x, y)$. For simplicity consider the binary case. Furthermore, since we restricted ourselves to loss functions applied on the top layer only, assume that $\epsilon$ is applied only to the weights of the top layer. In our setup, the Hessian $H$ is now restricted to the weights of the final layer. Since the $0 - 1$ loss is bounded above by the cross-entropy loss (times a factor $1/\log(2)$), we get

$$L_\sigma(w, x, y) \leq \frac{1}{\log(2)} \mathbb{E}_\epsilon \left[ L_c(w + \epsilon, x, y) \right] \approx \frac{L_c(w, x, y)}{\log(2)} + \frac{\sigma^2}{2\log(2)} \operatorname{Tr}(H). \tag{9}$$

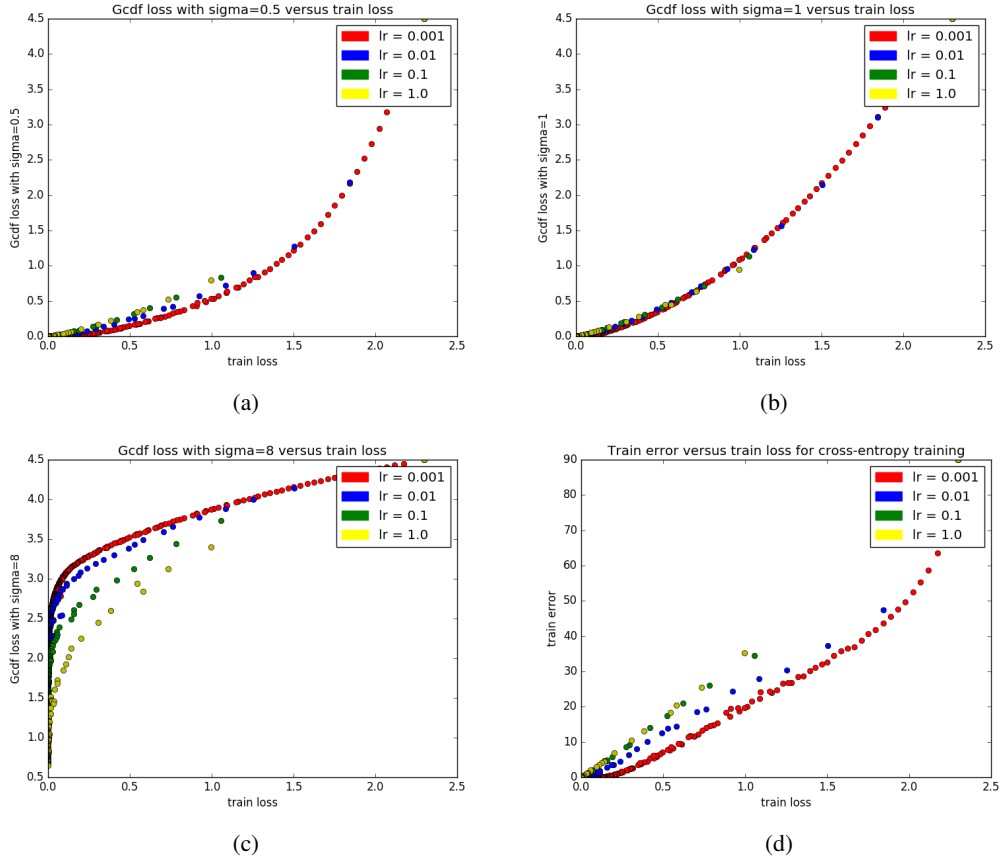

Figure 6: For each value of the training loss (cross-entropy) achieved during training on CIFAR10, we plot the Gcdf loss at that time on the y axis. In (a), we use a smaller value of $\sigma = 0.5$, in (b) an intermediate value of $\sigma = 1$ and in (c) a larger value of $\sigma = 8$. The train error is plotted against the train loss in (d). Four training runs corresponding to four different learning rates are drawn in each case.

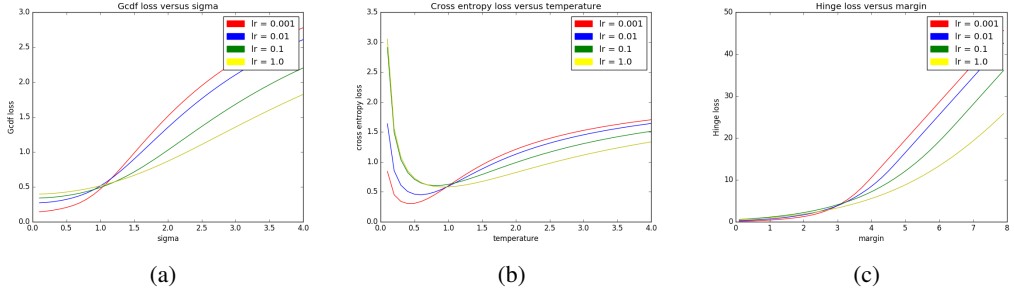

Figure 7: At a given fixed cross-entropy training loss (here approximately 0.6 in all cases), the Gcdf loss for varying $\sigma$'s in (a), the cross-entropy loss with varying temperatures in (b) and the Hinge loss with varying $\gamma$'s in (c) are plotted. Larger learning rates obtain better values of the alternative losses for larger $\sigma$'s, temperatures and $\gamma$'s while smaller learning rates are generally better for smaller values of these parameters.

Therefore, an optimization algorithm succeeding at finding a solution with small cross-entropy loss and small mean curvature of the cross-entropy loss must have a small Gcdf loss also. This might help to explain why larger learning rates and smaller batch sizes are good at implicitly minimizing

the Gcdf loss. Note however that this argument has some weaknesses. First, the approximation is only local and so might not be good for larger values of $\sigma$. Second, it cannot explain why smaller learning rates and larger batch sizes are better for smaller values of $\sigma$. Future work could concentrate on finding a rigorous explanation for these results.

Understanding the inductive biases of different optimization algorithms for training deep networks might allow to make the bias more explicit, that is to incorporate it in the loss function. We think that one strategy to make progress toward this long term goal might be to study how alternative loss functions are being implicitly minimized by a given optimization algorithm. This paper considered the learning rate and batch size parameters when training with SGD. A clear avenue for future research is to extend the investigation to adaptive first-order methods (which can sometimes exhibit worse generalization performance than SGD (Wilson et al., 2017)) and second-order methods.

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

## A    RESULTS ON MNIST

This section contains the results when training a 6-layer fully connected network with batch normalization on MNIST. No data augmentation is used. The optimization algorithm is SGD with momentum (0.9) and without weight decay. The learning rate is constant during all training.

## B    MORE RESULTS ON CIFAR10

This section contains additional results when training a convolutional network with batch normalization on CIFAR10. The network consists of two convolutional layers with max pooling followed by 3 fully connected layers. No data augmentation is used. The optimization algorithm is SGD with momentum (0.9) and without weight decay. The learning rate is constant during all training.

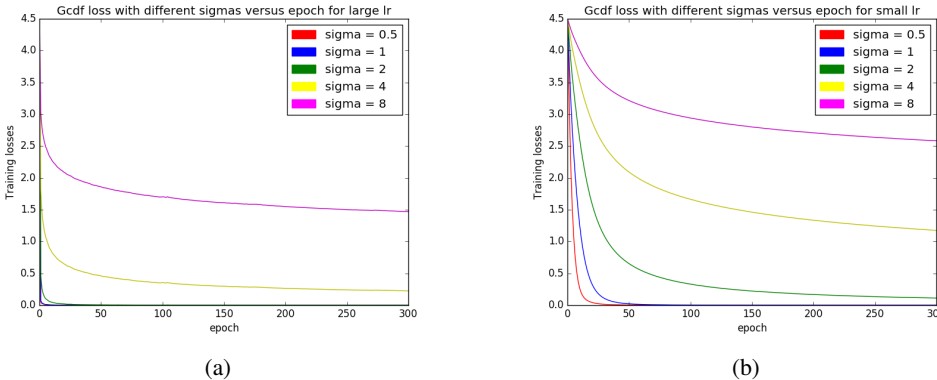

(a)                                        (b)

Figure 8: Tracking the Gcdf loss with different standard deviation parameters while training with the standard cross-entropy loss on MNIST. A relatively large learning rate of $0.1$ is used in (a) while a much smaller learning rate of $0.001$ is used in (b).

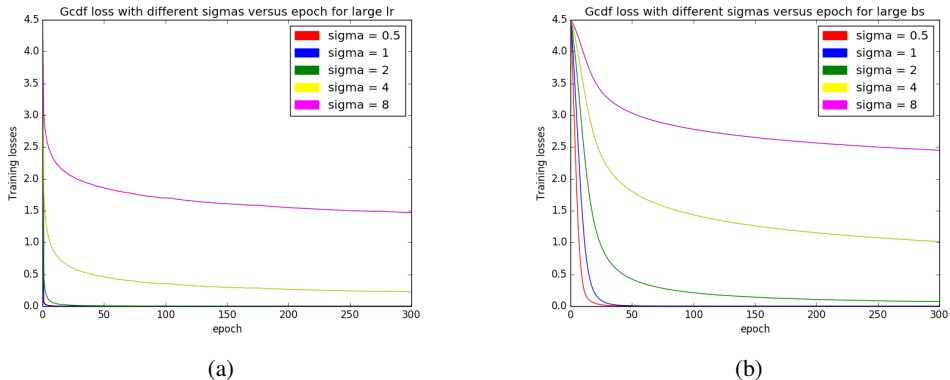

(a)                                        (b)

Figure 9: Tracking the Gcdf loss with different standard deviation parameters while training with the standard cross-entropy loss on MNIST. A relatively small batch size of $256$ is used in (a) while a much larger batch size of $16384$ is used in (b).

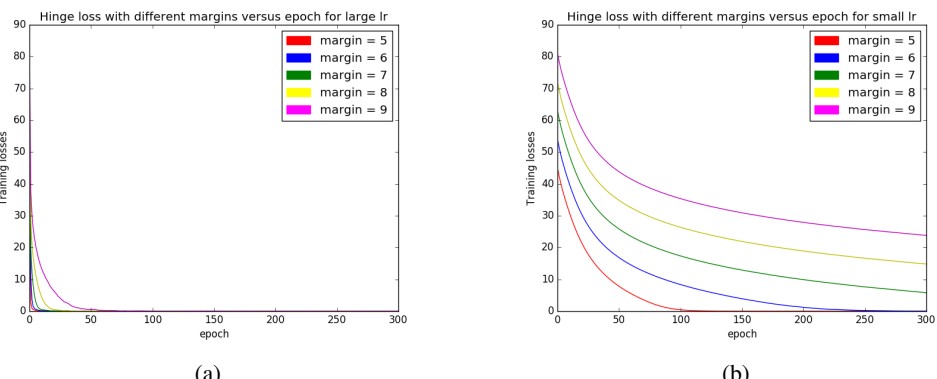

(a)                                        (b)

Figure 10: Tracking the Hinge loss with different margin parameters while training with the standard cross-entropy loss on MNIST. A relatively large learning rate of $0.1$ is used in (a) while a much smaller learning rate of $0.001$ is used in (b).

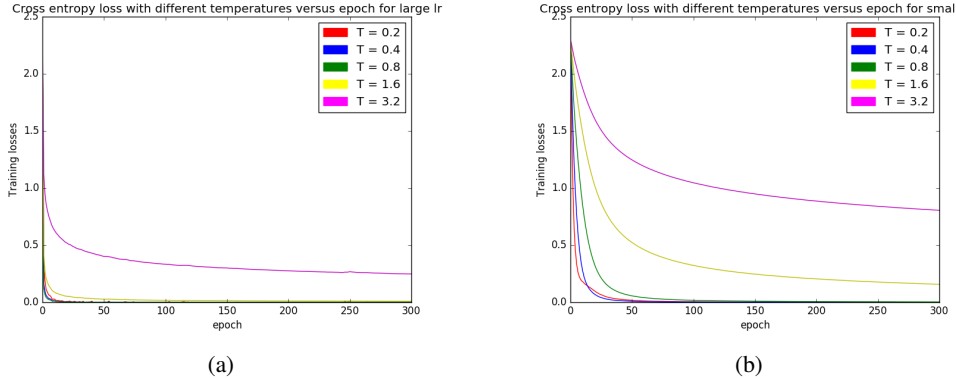

(a)            (b)

Figure 11: Tracking the the cross-entropy loss with different temperatures while training with the standard cross-entropy loss ($T = 1$) on MNIST. A relatively large learning rate of $0.1$ is used in (a) while a much smaller learning rate of $0.001$ is used in (b).

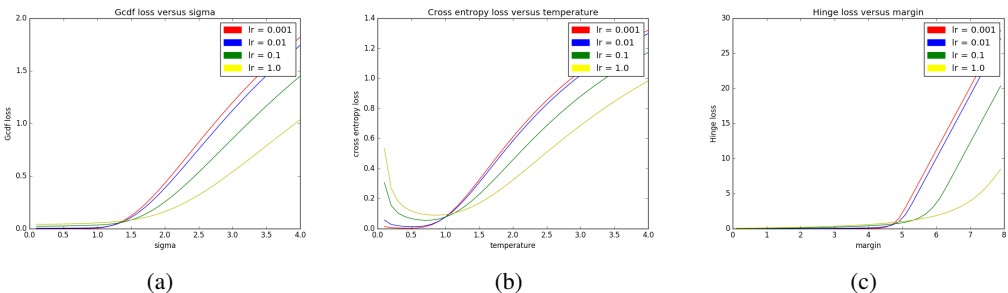

(a)          (b)          (c)

Figure 12: At a given fixed cross-entropy training loss (here approximately $0.08$ in all cases), the Gcdf loss for varying $\sigma$'s in (a), the cross-entropy loss with varying temperatures in (b) and the Hinge loss with varying $\gamma$'s in (c) are plotted. Larger learning rates obtain better values of the alternative losses for larger $\sigma$'s, temperatures and $\gamma$'s while smaller learning rates are generally better for smaller values of these parameters. The training dataset is MNIST.

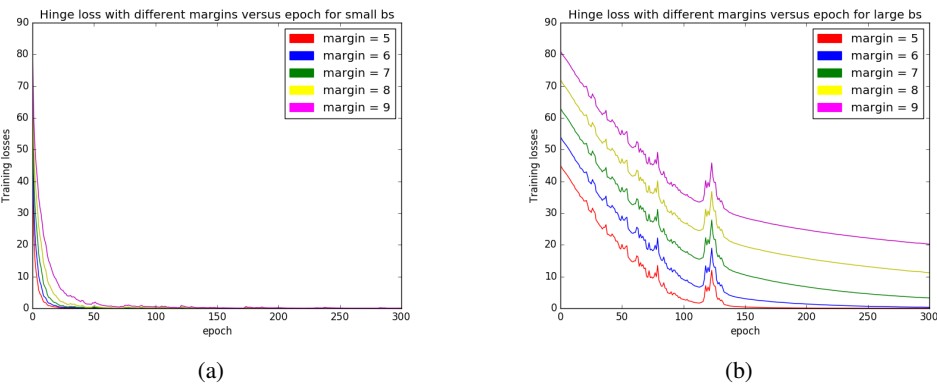

(a)            (b)

Figure 13: Tracking the Hinge loss with different margin parameters while training with the standard cross-entropy loss on CIFAR10. A relatively batch size of $256$ is used in (a) while a much larger batch size of $16384$ is used in (b). Even though both batch sizes succeed at minimizing to almost zero the loss function they are trained on, the larger batch size does not implicitly minimize as well the Hinge loss for larger margins.

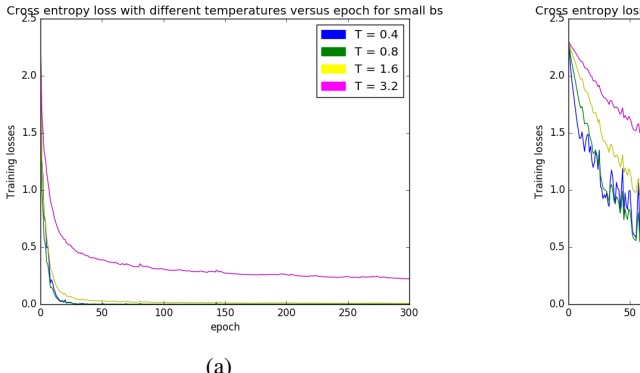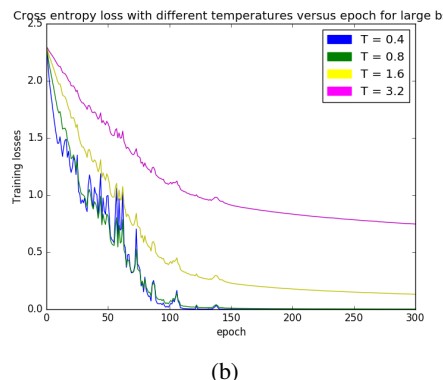

Figure 14: Tracking the cross-entropy loss with different temperature parameters while training with the standard cross-entropy loss ($T = 1$) on CIFAR10. A relatively batch size of $256$ is used in (a) while a much larger batch size of $16384$ is used in (b). Even though both batch sizes succeed at minimizing to almost zero the loss function they are trained on, the larger batch size does not implicitly minimize as well the cross-entropy loss for larger temperatures.

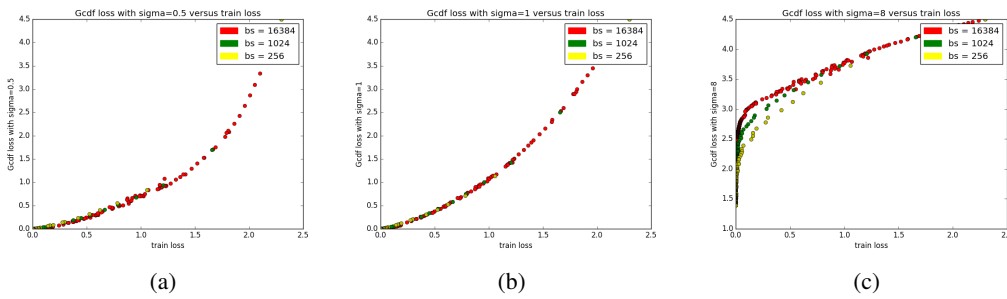

Figure 15: For each value of the training loss (cross-entropy) achieved during training on CIFAR10, we plot the Gcdf loss at that time on the y axis. In (a), we use a smaller value of $\sigma = 0.5$, in (b) an intermediate value of $\sigma = 1$ and in (c) a larger value of $\sigma = 8$. Three training runs corresponding to three different batch sizes are drawn in each case. Larger batch sizes are slightly better at minimizing the Gcdf loss with small $\sigma$ during training while smaller batch sizes are better at minimizing the Gcdf loss with larger $\sigma$. There exists an intermediate value (here $\sigma = 1$) where all the batch sizes considered in our experiments are essentially equivalent at implicitly minimizing the Gcdf loss.

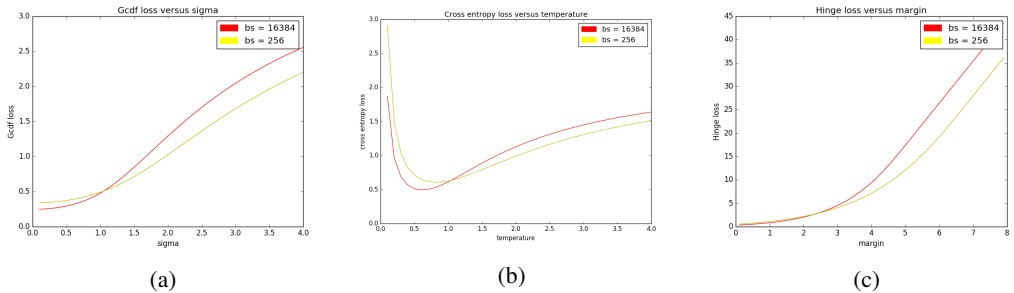

Figure 16: At a given fixed cross-entropy training loss (here approximately $0.6$ in all cases), the Gcdf loss for varying $\sigma$'s in (a), the cross-entropy loss with varying temperatures in (b) and the Hinge loss with varying $\gamma$'s in (c) are plotted. The small batch size obtain better values of the alternative losses for larger $\sigma$'s, temperatures and $\gamma$'s while the large batch size is better for smaller values of these parameters. The training dataset is CIFAR10.

