# OpenReview forum: "On the implicit minimization of alternative loss functions when training deep networks"
_ICLR.cc/2020/Conference — Reject_

### Official Review · AnonReviewer3 · 2019-10-10
**Official Blind Review #3**

**Rating:** 1

**Review:**

This paper want to show that minimizing cross-entropy loss will simultaneously minimize Hinge loss with different margins, cross-entropy loss with different temperatures and a newly introduced Gcdf loss with different standard deviations. The main contribution is a new gcdf loss based on Gaussian-perturbed parameters. However, this loss can only be used with linear models. For deep models, the authors suggest that only measure this loss on the top layer of model.

The motivation is week. Seems most of these loss functions only depend on s_i - s_j, the difference between logits. And the optimization with cross-entropy loss wants to maximize this difference between logits corresponding to true labels and false labels, which is obviously minimize difference loss functions. So I do not feel surprise that optimizing the neural network with cross-entropy loss will minimize other kinds of losses.

The format is poor. The figures occupy most of the places and thus let me feel the contents of this paper is somewhat weak.

Detailed Comments:
1. In Sec 3.1, it will be better to mention that y belongs to +1 and -1. Also, in the last sentence in Sec 3.1, the meaning of x is normalized is ambiguous. I guess the authors want to say if x is unit norm?
2. Will adding the regularization of the feature map norm violate the performance?
3. What do the authors want to say in Sec 4.3? Does the relation of learning rate and convergence rate of gcdf loss indicate some non-trivial results?
4. I cannot understand the meaning of Figure 7. Obviously different learning rate may lead to different training process and thus the different solution and different s_i - s_j. But I think this is not related to the different losses. With simple calculation I think we can find all of these losses have some relation with s_i - s_j and thus we can directly say different learning rate lead to different s_i - s_j and there is no need to relate s_i - s_j to these losses.

Overall I find the claims of this paper is somewhat weak. Gcdf loss seems related to some kinds of adversarial robustness that can be individual interest, but the current paper is still far from the standard of publication. Some more interesting and valuable directions can be the theoretically analysis of the equivalence between the objective optimization, e.g. minimizing cross-entropy loss is equivalent to minimizing gcdf loss for example, as well as the empirical comparison between the different losses, like selecting some losses as objective can boost the performance on some aspects.

**Experience Assessment:**

I have read many papers in this area.

**Review Assessment: Checking Correctness Of Derivations And Theory:**

I carefully checked the derivations and theory.

**Review Assessment: Checking Correctness Of Experiments:**

I did not assess the experiments.

**Review Assessment: Thoroughness In Paper Reading:**

I read the paper thoroughly.

---

### Official Review · AnonReviewer1 · 2019-10-22
**Official Blind Review #1**

**Rating:** 3

**Review:**

This paper adds to a large body of research on implicit regularization:
in the context of deep networks, that, when SGD with a certain step
size or batch size is used, from among parameter vectors that fit
the data equally well, some are much more likely to be chosen than
others.  This paper attacks this question through the lens of
loss functions: when SGD is applied to the usual softmax loss
with a given learning rate, how does the choice of learning
rate effect how rapidly other loss functions are reduced?  They
pay special attention to a loss function that they call Gcdf,
which is motivated by "wide minima" considerations.  In particular,
for the Gcdf to be small, not only must training examples be
classified correctly, but randomly perturbing the weights in
the last layer should not change this correct prediction.

I am convinced that Figures 6 and 12 of this paper show
that optimization of the softmax with a larger step size
implicitly optimizes a loss function that rewards robustness
to a greater extent than when a small step size is used.
I find this interesting.

The authors do a nice job of summarizing a lot of related work.

The Gcdf loss is similar to the ramp loss used in [1]
(see Section 3.1) and elsewhere, including to analyze
generalization in deep learning.  It is also like the
potential function optimized by RobustBoost
(see (4) of [2]) -- the RobustBoost loss function does
not scale by the norm of x, but since it is an ensemble
method, the role of x is played by the predictions of
members of the ensemble, which have a fixed scale.

I assume that, when they evaluate the Gcdf loss for a deep
network, they normalize by the norm of the last hidden
layer.  If this is true, it only captures "wide minima"
in the sense of being robust with respect to perturbations
of the output layer.  (They seem to acknowledge this point
in their paper.)

The results in the paper are not described in enough
detail to be reproduced.  For example, I don't see
where they specify the architecture of the network
that they used in Section 4.

The experiments are limited and narrow in scope.

In Figures 2 and 3 I don't see that they have adequately controlled
for the effect of the learning rate on how fast the explicitly
minimized loss is reduced.  Part of the effect observed is simply
that, when a small learning rate is used, after a given number of
epochs, the weights are just not changed much, so that no loss is
reduced much.  In Figure 2, I find it strange that they did not plot
the values for T=1.

Figure 6 is the most interesting to me.  It seems to show that
training the same loss function with a larger learning rate
effectively optimizes the gcdf loss that rewards sacrificing
training error to achieve stronger robustness.

Small point: the first time I read Appendix A, I thought that the
results were not there.  It was only later that I saw the figures with
the MNIST results.  It would be helpful if the authors wrote
"The results are in Figures 8-12".

While, as I wrote above, I did find Figure 6 interesting, I feel that
the increment of this research over the large body of previous work
on this topic is not enough to justify publication in ICLR.




[1] Bartlett, Peter L., Dylan J. Foster, and Matus
J. Telgarsky. "Spectrally-normalized margin bounds for neural
networks." Advances in Neural Information Processing Systems. 2017.

[2] https://arxiv.org/pdf/0905.2138.pdf


**Experience Assessment:**

I have read many papers in this area.

**Review Assessment: Checking Correctness Of Derivations And Theory:**

N/A

**Review Assessment: Checking Correctness Of Experiments:**

I carefully checked the experiments.

**Review Assessment: Thoroughness In Paper Reading:**

I read the paper thoroughly.

---

### Official Review · AnonReviewer2 · 2019-10-23
**Official Blind Review #2**

**Rating:** 3

**Review:**

This paper makes a step towards understanding of the implicit bias of optimization algorithms in deep learning. The authors consider alternative loss functions for deep networks: (1) the temperature-scaled cross-entropy loss with different values of the temperature; (2) the hinge-loss with different values of the margin parameter; (3) the Gcdf loss with different values of the variance parameter. The paper introduces the Gcdf loss which is derived as a modification of the 0-1 loss under the noise in the parameters of the linear output layer. The authors propose to use the alternative losses as measures of margin and sharpness associated with a solution found by an optimization algorithm. The experiments show how SGD in different learning scenarios (low/high learning rate and small/large batch) performs implicit minimization of the alternative loss functions with different parameters. Specifically, using larger learning rates/smaller batch sizes is shown to implicitly minimize the losses corresponding to higher values of the temperature/margin/variance. The results provide insights about margins and sharpness of solutions found by different modes of SGD.

The direction explored in the paper is important for the understanding of the connections between optimization, properties of the loss landscapes (such as sharpness), and generalization. The results reported in the paper are interesting. However, currently I am not convinced that the contributions are sufficient for publication at ICLR as the scope of the performed analysis is limited. In my view, the study is not comprehensive enough and the paper would benefit from incorporating additional results.

Detailed comments:


1) My main concern is that currently there is very little explanation provided for the observed experimental findings. The paper would strongly benefit from additional results focused on identification and verification of the mechanisms behind the observed behavior of the optimizer.

2) Many connections mentioned in the paper are left unexplored. It would help to investigate the mentioned connections between the implicit minimization of the considered losses and sharpness, curvature, and generalization. A similar design of the experiment can be used in which the alternative loss values can be tracked alongside with the validation loss (or multiple losses) as well as the measures of sharpness and the characteristics of the Hessian.

3) Another direction for improvement is the extension of the set of analyzed settings (as it was mentioned in the discussion section). This includes performing the analysis for a broader set of architectures (potentially with different normalization schemes), optimizers, and choices of the hyperparameters (momentum, weight decay). These experiments would help to better understand the observed phenomenon and analyze the effect of different settings.



**Experience Assessment:**

I have published one or two papers in this area.

**Review Assessment: Checking Correctness Of Derivations And Theory:**

N/A

**Review Assessment: Checking Correctness Of Experiments:**

I carefully checked the experiments.

**Review Assessment: Thoroughness In Paper Reading:**

I read the paper thoroughly.

---

### Decision · Program_Chairs · 2019-12-19

**Decision:**

Reject

**Comment:**

The paper proposes an interesting setting in which the effect of different optimization parameters on the loss function is analyzed.  The analysis is based on considering cross-entropy loss with different softmax parameters, or hinge loss with different margin parameters.  The observations are interesting but ultimately the reviewers felt that the experimental results were not sufficient to warrant publication at ICLR.  The reviews unanimously recommended rejection, and no rebuttal was provided.